# Experimental Study of Downburst Wind Flow over a Typical Three-Dimensional Hill

Yong Chen [1], Yuhan Li [1], Jianfeng Yao [2], Guohui Shen [1,*] , Wenjuan Lou [1], Haiwei Xu [1] and Yong Guo [3]

1   College of Civil Engineering and Architecture, Zhejiang University, Hangzhou 310058, China;
    cecheny@zju.edu.cn (Y.C.); 21912164@zju.edu.cn (Y.L.); louwj@zju.edu.cn (W.L.); haiwei163@163.com (H.X.)
2   College of Civil Engineering and Architecture, Zhejiang University of Water Resources and Electric Power,
    Hangzhou 310018, China; yaojf@zjweu.edu.cn
3   China Energy Engineering Group Zhejiang Electric Power Design Institute Co., Ltd.,
    Hangzhou 310012, China; buyage@126.com
*   Correspondence: ghshen@zju.edu.cn; Tel.: +86-0571-88208737

**Abstract:** To achieve a comprehensive understanding of a three-dimensional (3D) wind field and the speed-up phenomenon in a downburst wind flow over a 3D hilly terrain, a succession of laboratory tests utilizing 12 hill models with cosine-squared cross-section was conducted using a physical downburst simulator with a jet diameter of 0.6 m. By placing the models in the strong horizontal wind region and the strong vertical wind region, the corresponding wind profiles for both the horizontal and vertical velocities were measured. It was found that the wind flowed predominantly over the crest of the hill in the case of low hills, whereas wind flow around the hill body became increasingly pronounced as the hill height increased. In addition, the speed-up region, where the horizontal wind velocity exceeds the impinging jet velocity, was identified, and found to move from the crest to the two sides of the hill as the hill height increased. Accordingly, the most significant topographic multipliers of all locations on the hill might appear at the crest, the hill foot, or elsewhere, depending largely on the hill height. Among all cases, the maximum topographic multiplier was 1.12, and occurred at the ridge, while the ratio of hill height to jet height was 5/12. Additionally, empirical equations are presented to facilitate the determination of wind loads induced by a downburst flow over an isolated hill.

**Keywords:** downburst; hilly terrain; wind-tunnel test; three-dimensional wind field; topographic multiplier

## 1. Introduction

Downbursts, resulting in a high near-ground wind velocity, are recognized as a primary factor in most transmission tower collapse events [1,2], and have received much attention in wind engineering. Figure 1 shows several collapsed transmission towers, erected on hilly terrain, due to a severe downburst event in Shaoxing, China. In addition, concerns regarding downburst-induced wind loadings on other structures that are vulnerable to winds, e.g., long-span bridges [3] and rail overhead lines [4], have also been raised recently. Considering downburst effects on structures erected on islands, hills, or mountains, a large number of studies [5–13] utilizing various idealized hill models have been conducted in the past decades, and have reached a consensus that the aerodynamic interference of hilly terrain leads to a speed-up phenomenon, which a priori increases the risk of structural failure. Nonetheless, Abd-Elaal et al. [5] recently emphasized that the effects of topography on downburst wind fields require further research, as most studies have been limited to two-dimensional (2D) topographic features, and the changes in the vertical downburst wind speed component are not well understood.

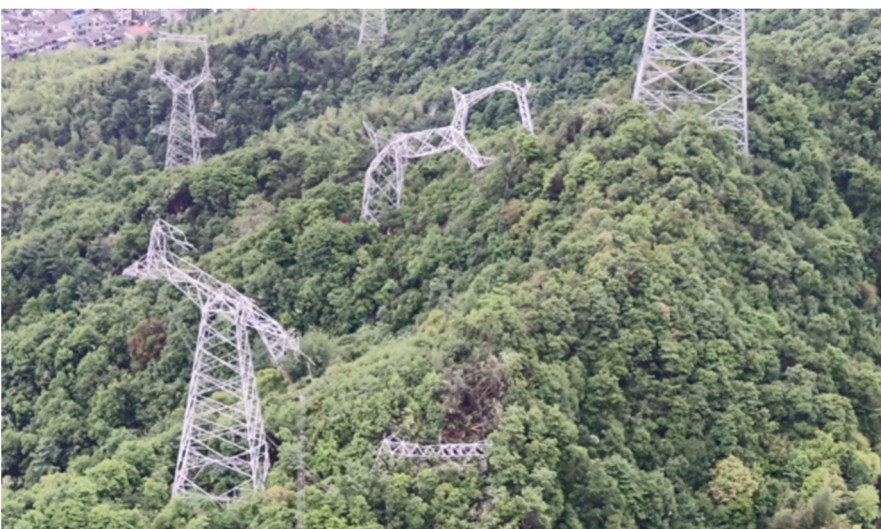

**Figure 1.** Collapses of transmission towers following a downburst.

To the best of the authors' knowledge, no field measurement-based study has been performed to examine the effects of hilly terrain on a downburst wind field from the perspective of wind engineering. Moreover, in terms of wind profile, which is a factor in determining wind loadings on structures, the relevant field test results are limited and scattered in the literature. The earliest study of downburst phenomena can be traced back to the pioneering work of Fujita [14,15], who named the downburst and reported an impressive wind record in a downburst event near Washington, DC, during which the peak velocity reached 67 m/s at 4.9 m above the ground [16]. Choi [17] investigated variation in wind velocity with height during a thunderstorm event in Singapore, and documented that the measured wind profiles matched the results from the impinging jet model. Using the data of the Joint Airport Wind Shear (JAWS) Project, Hjelmfelt [18] presented the ring vortex structure of the downburst flow, and pointed out that the outflow was found to resemble the impinging jet model in terms of wind profiles. Chen and Letchford [19] reported the full-scale vertical profiles of full-scale nonstationary downburst wind speeds via seven evenly deployed towers at the Reese Technology Center, Texas, USA, and found that the time function for the time-varying mean speed is significantly different from event to event. Additionally, using data collected via Doppler radars, Gunter and Schroeder [20] showed the evolution of the mean wind profile from a uniform to an impinging jet shape.

In contrast, substantial laboratory tests and numerical simulations, most of which are based on an impinging jet model, have been conducted to reveal insights into hilly terrain-disturbed downburst wind fields. Note that the impinging jet model conventionally employed for these studies makes it difficult to reproduce the dynamic characteristics of downbursts, particularly the ring vortex [16]. Selected experimental results and numerical simulation results from the literature are tabulated in Table 1, where $D_{jet}$ and $h$ are the jet diameter and the jet height of the downburst, respectively; $H$, and $\phi = H/2L_u$ are the height and the representative slope of the terrain, respectively; $L_u$ is the horizontal distance from the crest to the site with altitude of $H/2$; $r_h$ is the horizontal distance from the crest to the stagnation point of the downburst; and $M_{t,max}$ is the topographic multiplier for the maximum velocity amplification [9,10]. In the table, for the studies in which the maximum of $M_{t,max}$ is not declared, the data are read directly from the relevant figures, which may lead to the existence of a small inevitable error. The studies show that the speed-up phenomenon becomes increasingly significant as the hill height decreases. In addition, most studies were concerned with 2D terrains, and the scattered research examining three-dimensional (3D) terrain were limited to measuring the wind fields along ridges in the radial direction, which might lead to an underestimation of the 3D effects of the terrain, namely the maximum speed-up, which may also occur on the sides of the hill, if the hill is

sufficiently high. Moreover, studies of the changes in the vertical wind velocity are rather rare, even though vertical wind, which can lead to a skewed wind effect on structures, features in the downburst wind field. Recently, Abd-Elaal et al. [5] stated that hilly terrain-disturbed vertical wind could reach half the impinging jet speed, and there is a need to investigate transmission-line systems subjected to vertical winds.

**Table 1.** Selected topographic multipliers in the literature.

| Reference | $D_{jet}$ | $h/D_{jet}$ | Terrain ** | $\phi$ | $H/D_{jet}$ | $r_h/D_{jet}$ | Maximum of $M_{t,max}$ | | Method |
|---|---|---|---|---|---|---|---|---|---|
| | | | | | | | Value | Location | |
| Selvam et al. [6] | 900 m | 2.98 | E (2D) | 0.25 | 0.044 | 1.29 | / | Crest | CFD |
| Wood et al. [8] | 0.31 m | 2.00 | E (2D) | 0.5 | 0.129 | 1.5, 2.0 | 0.96–1.22 * | Crest | CFD |
| Mason et al. [9] | 0.104 m | 2.00 | E, TH, BH (2D) | 0.2, 0.5 | 0.024,0.048 | 1.0, 1.5 | 1.09–1.26 | Crest | CFD |
| Mason et al. [10] | 1000 m, 3000 m | 0.67, 2.00 | BH, BE (2D) | 0.2–1.0 | 0.017- 0.100 | 0.75–1.75 | 1.00–1.71 | Crest | CFD |
| Abd-Elaal et al. [5] | 0.75 m | 3.50 | E, TH (2D) | 0.2 | 0.048 | 1.74 | 1.19, 1.21 | Crest | CFD |
| Fang et al. [12] | 0.60 m | 2.00 | E(3D) | 0.25–0.5 | 0.125–0.250 | 1.5 | 0.9–1.0 * | Crest | CFD |
| Letchford et al. [7] | 1.225 m | 0.72 | E (3D) | 0.2–0.6 | 0.082 | 1.0, 1.4 | 1.08–1.33 * | Crest | Test |
| Wood et al. [8] | 0.31 m | 2.00 | E (3D) | 0.5 | 0.129 | 1.5, 2.0 | 0.98–1.11 * | Crest | Test |
| Mason et al. [9] | 0.104 m | 2.00 | E, TH, BH (3D) | 0.2, 0.5 | 0.048 | 1.0 | 1.10–1.24 | Crest | Test |
| Ji et al. [11] | 0.40 m | 3.00 | CH (3D) | 0.58 | 0.5 | 1.8, 2.8 | 0.80 *, 1.02 * | Hill foot | Test |
| Fang et al. [12] | 0.60 m | 2.00 | E (3D) | 0.25–0.5 | 0.125–0.250 | 1.5 | 0.9–1.1 * | Crest | Test |

\* $M_{t,max}$ is computed via the data read directly from the relevant figures. ** "E", "BE", "BH", "TH", and "CH" indicate that the terrains investigated were escarpment, bell-shaped escarpment, bell-shaped hill, triangular hill, and cosine-shaped hill respectively.

In this paper, a succession of 3D hill models with a cosine-squared cross-section are employed for laboratory tests using a physical downburst simulator. The hilly terrain-disturbed downburst wind field is measured, in terms of the horizontal and the vertical wind velocities. By using the test results of a downburst wind flow over a flat surface, the spatial distributions of the topographic multiplier for the maximum horizontal wind velocity are investigated, in terms of contour maps. Accordingly, the effects of the hill size and the hill position on the wind profiles and the speed-up region are analyzed. Finally, for the benefit of future structure design, empirical formulas are presented that enable the calculation of wind loads induced by horizontal and the vertical winds, respectively, taking into account the location of the structure, and the effects of the hill features.

## 2. Experimental Setup

As shown in Figure 2a, the laboratory tests were conducted by using a physical downburst simulator in Zhejiang University. The impinging jet consists of four parts: a fan, a diffuser, a settling chamber, and a contraction unit [21]. In the tests, the jet diameter $D_{jet}$ was 0.6 m; the jet height $h$, from the ground to the nozzle, was set to be two times the jet diameter; $r$ denotes the radial horizontal distance from the stagnation point. The temperature of the downdraft was the same as the indoor ambient temperature, ranging from 6 to 14 °C. A Cartesian coordinate system for the hill was employed and is illustrated in Figure 2b, with the origin set at the center of the hill bottom face, and where $x$ and $y$ are the orthogonal horizontal coordinates. Without loss of generality, the $x$-axis of the hill coincided with the radial direction of the jet flow. A cosine-squared cross-section was adopted to represent an idealized axisymmetric hill [22], in the form of

$$z(x,y) = \begin{cases} H\cos^2(\pi\sqrt{x^2+y^2}/2L), & \sqrt{x^2+y^2} < L \\ 0, & \sqrt{x^2+y^2} \geq L \end{cases} \tag{1}$$

where $L$ is the radius of the bottom face of the hill and $z$ is the altitude of the point on the hill surface. Accordingly, Equation (1) yields $L = 2L_u$. Moreover, a local coordinate, $Z$, is used to denote the relative height from the surface of the hill. A total of 12 hill models, with parameters tabulated in Table 2, were employed in the laboratory tests to investigate the effects of the height, size, and location of the hill on the wind field above the hill. The hill models can be classified into two categories, namely the hill models with $L = 0.5D_{jet}$ and those with $L = 1.0D_{jet}$. The hill models were made from rigid extruded polystyrene foam

board. Assuming a scale of 1:1000 and an average vegetation height of 2.0 m, the vegetation was simulated by gluing 2 mm long plastic fibers onto the surface of the hill models (see Figure 3), according to previous work by Shen et al. [22], and the surface roughness of the hill models was approximately the same for all models. Figure 3 shows the hill models with the simulated vegetation. To examine the strong horizontal wind region, the hill models were first placed at three locations for the experimental study, namely $r_h = 0.8D_{jet}$, $1.0D_{jet}$, and $1.2D_{jet}$. For the next part of the test, the hill models were placed at $r_h = 0.0D_{jet}$, where a strong vertical wind would occur.

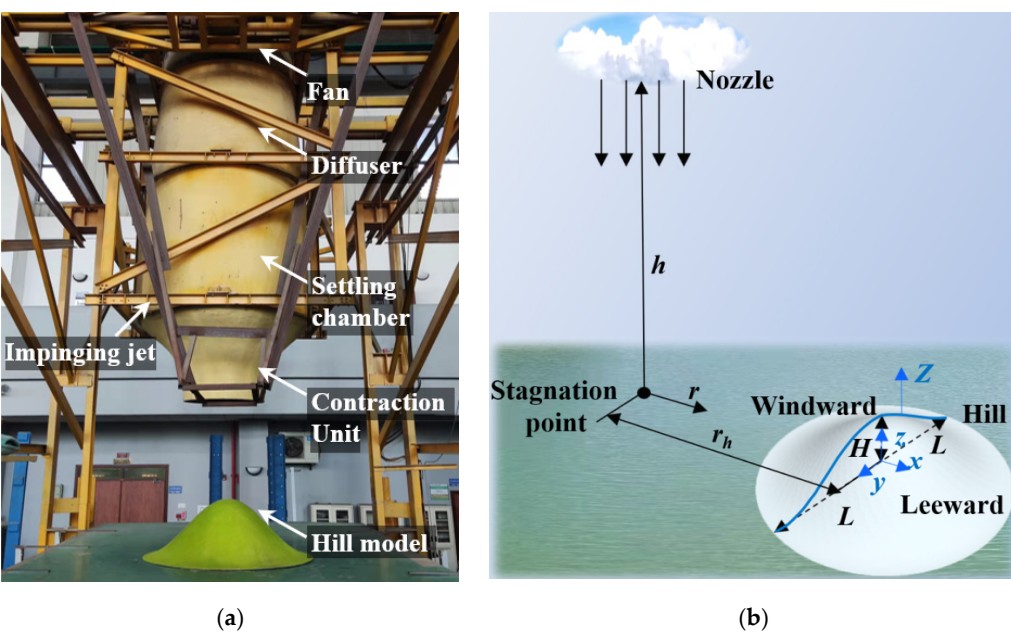

(**a**)                                                    (**b**)

**Figure 2.** Experimental setup (**a**) and coordinate systems (**b**).

**Table 2.** Dimensions of hill models.

| Hill Model | H1 | H2 | H3 | H4 | H5 | H6 | H7 | H8 | H9 | H10 | H11 | H12 |
|---|---|---|---|---|---|---|---|---|---|---|---|---|
| $H$ (m) | 0.1 | 0.2 | 0.3 | 0.4 | 0.5 | 0.6 | 0.1 | 0.2 | 0.3 | 0.4 | 0.5 | 0.6 |
| $L$ (m) | 0.3 | 0.3 | 0.3 | 0.3 | 0.3 | 0.3 | 0.6 | 0.6 | 0.6 | 0.6 | 0.6 | 0.6 |
| $H/L$ | 1/3 | 2/3 | 1.0 | 4/3 | 5/3 | 2.0 | 1/6 | 1/3 | 1/2 | 2/3 | 5/6 | 1.0 |

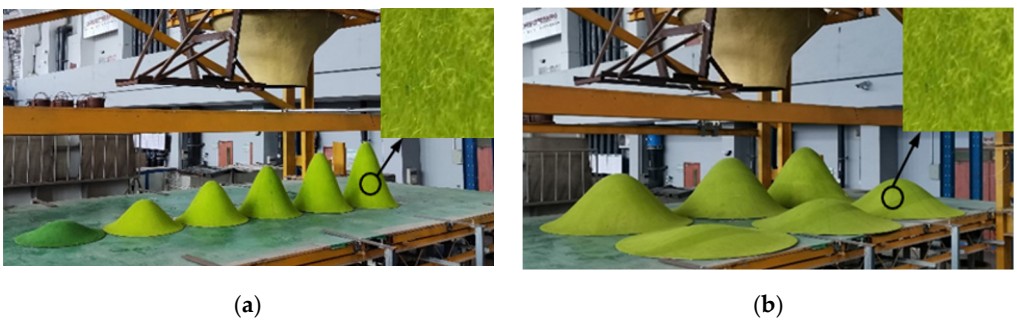

(**a**)                                                    (**b**)

**Figure 3.** Photos of hill models: (**a**) $L = 0.5D_{jet}$; (**b**) $L = 1.0D_{jet}$.

The Cobra Probe [23] produced by Turbulent Flow Instrumentation Pty Ltd. was employed to measure the wind velocity field of the downburst. The Cobra Probe is a 4-hole pressure probe that provides dynamic, 3-component and local pressure measurements in real-time. The Cobra Probe features a linear frequency-response from 0 Hz (mean flow) to more than 2000 Hz and is available in various ranges for use between 2 m/s and 100 m/s. In addition, 3-component velocity and static pressure measurements within a

± 45° acceptance cone can be performed. The measurement accuracies of the velocity and the flow angle are ±0.3 m/s and ±1°, respectively.

Taking advantage of symmetry, the wind velocity profiles at 21 measuring locations were measured. Figure 4a shows the measuring locations at the hill foot (0.0*H*), ridge (0.25*H*, 0.5*H*, 0.75*H*), and crest (1.0*H*), respectively. In this study, the ridge is defined as the profile of the hill model, where the crest and the hill foot are excluded, as shown in Figure 4b. The measuring locations were arranged at the ridges of the *x*-axis, *y*-axis, 45° ray, and 135° ray, respectively, namely at the ridges whose projections on the x–y plane coincide with the *x*-axis, *y*-axis, 45° ray, and 135° ray respectively. The 45° ray and 135° ray are the closed half-lines in the x–y plane, whose initial points are at the origin of the coordinate system for the hill, having the clockwise angles of 45° and 135° respectively from the positive direction of the *x*-axis. For each measuring location, the arrangement of the measuring points in a vertical line were the same. For illustration, the relative heights of the measuring points for P2 are shown in Figure 4b.

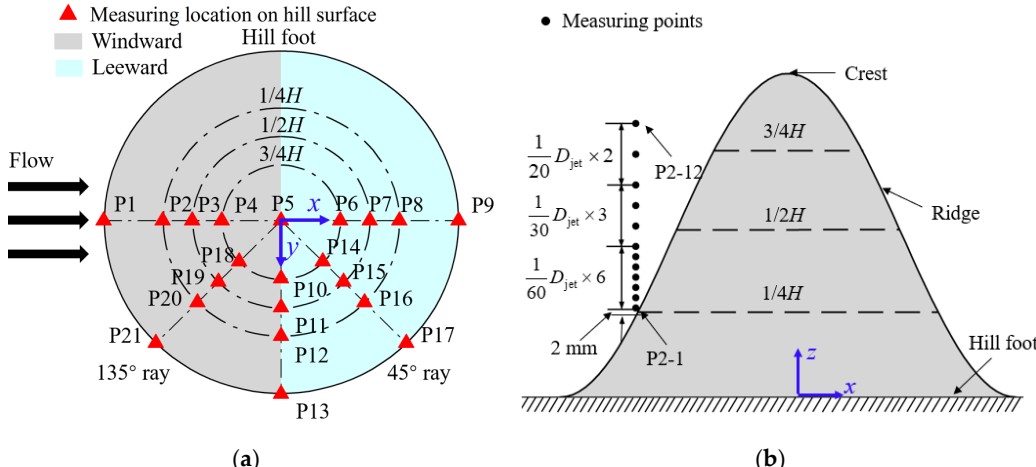

**Figure 4.** Layout of measuring locations and measuring points: (**a**) measuring location; (**b**) measuring points for measuring location of P2.

To determine the jet velocity for the tests, a preliminary study on the effects of the Reynolds number (Re) was performed in advance. The Reynolds number can be computed according to the formula Re = $V_{jet}D/\nu$, where $V_{jet}$ is the jet velocity, $D$ is the characteristic length equal to the radius of the curvature at the crest, and $\nu = 1.45 \times 10^{-5}$ m$^2$/s is the kinetic air viscosity at 15 °C. Four hill models, namely H1, H6, H7, and H12 were selected for the preliminary study. The hill models were placed at the position of $r_h = 1.0D_{jet}$, and the jet velocity was varied as follows: $V_{jet}$= 4, 6, 8, 10, and 12 m/s. Figure 5 shows the corresponding wind profiles of the horizontal wind velocity at the crest, as well as those at the hill foot lying on the 135° ray. Similarly, the hill models were placed at the position of $r_h = 0.0D_{jet}$ and the jet velocity was varied, as described above. Figure 6 depicts the corresponding profiles of the vertical wind velocity at the crest. Note that any measured velocities of less than 2 m/s might be inaccurate, due to the limited sensitivity of the Cobra Probe. For example, in Figure 6, the points where $V_h/V_{jet} < 0.2$ on a curve with a jet velocity of 10 m/s might be questionable. It was found that for the same hill model, the variation in the Reynolds number due to changes in the jet velocity had negligible effects on both the horizontal and the vertical wind velocities, since the wind profiles almost coincide. Accordingly, considering both the requirements of the Cobra Probe and the capacity of the downburst simulator, $V_{jet}$= 12 m/s was employed in the following tests, and the corresponding Reynolds numbers are summarized in Table 3.

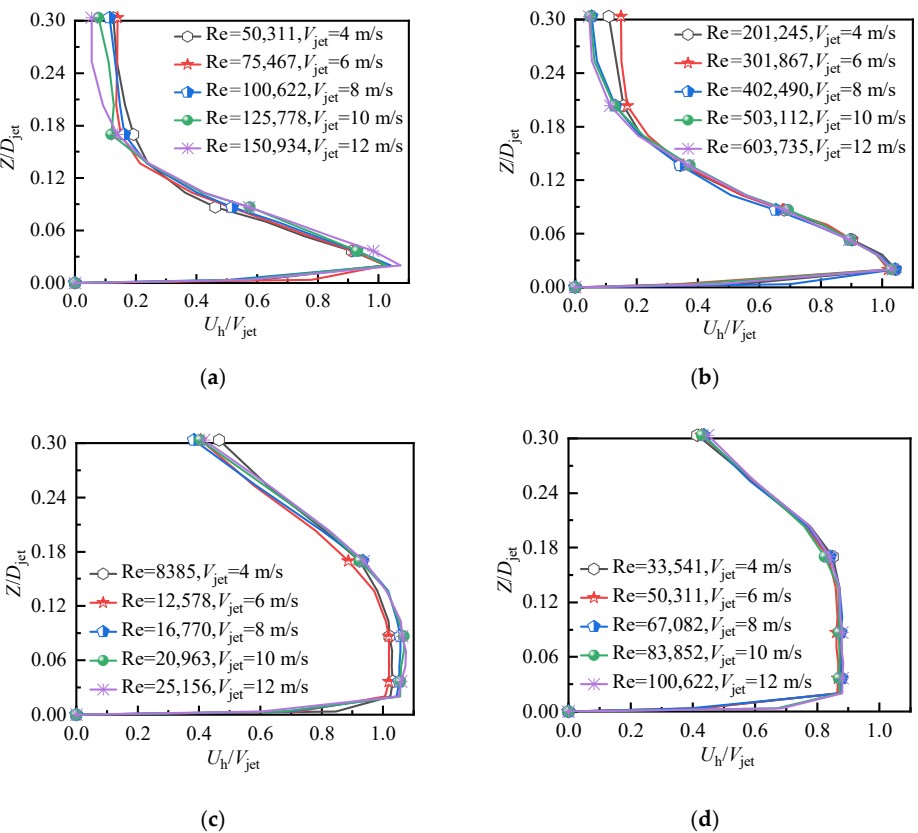

**Figure 5.** Profiles of horizontal velocity: (**a**) H1, crest; (**b**) H7, crest; (**c**) H6, hill foot on 135° ray; (**d**) H12, hill foot on 135° ray.

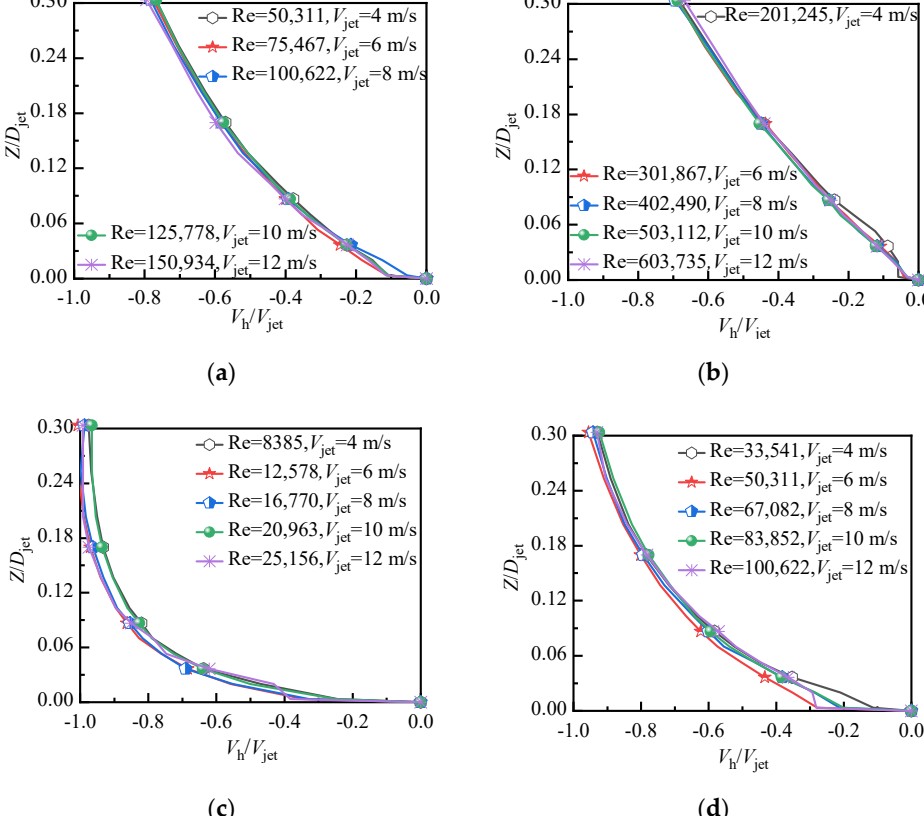

**Figure 6.** Profiles of vertical velocity: (**a**) H1, crest; (**b**) H7, crest; (**c**) H6, crest; (**d**) H12, crest.

**Table 3.** Reynolds numbers for hill models ($V_{jet}$ = 12 m/s).

| Hill Model | H1 | H2 | H3 | H4 | H5 | H6 | H7 | H8 | H9 | H10 | H11 | H12 |
|---|---|---|---|---|---|---|---|---|---|---|---|---|
| Re | 150,934 | 75,467 | 50,311 | 37,733 | 30,187 | 25,156 | 603,735 | 301,867 | 201,245 | 150,934 | 120,747 | 100,622 |

## 3. Results and Discussion

To facilitate the design of structures erected on a hill, the resultant horizontal wind velocity, *U*, as a function of *Z*, is herein utilized for the analysis of the wind field. To characterize the speed-up phenomenon, the topographic multiplier for the maximum velocity amplification [9,10], $M_{t,max}$, is utilized herein, in the form of

$$M_{t,max} = \frac{\max[U_{topography}(Z)]}{\max[U_{flat}(Z)]} \tag{2}$$

where the subscripts "topography" and "flat" represent the hilly terrain and the flat surface, respectively. The test results for the downburst wind flow over a flat surface without vegetation (undisturbed downburst, see Figure 7) show that the maximum $U_{flat}$ is about $1.0V_{jet}$, which is similar to the findings of previous studies [24–28]. The $M_{t,max}$ values greater than 1.0 indicate a strong horizontal wind occurring at that location with an intensity greater than the maximum found during an undisturbed downburst. Note that the traditional speed-up ratio, which is defined as the ratio of the wind velocity of a hilly terrain-disturbed atmospheric boundary layer (ABL) wind to that of an undisturbed ABL wind at the same height, cannot account for the speed-up of a downburst wind flow over hilly terrains, because the wind profiles of the undisturbed downburst wind vary with the distance from the stagnation point, in contrast to the invariant wind profile in the undisturbed ABL wind.

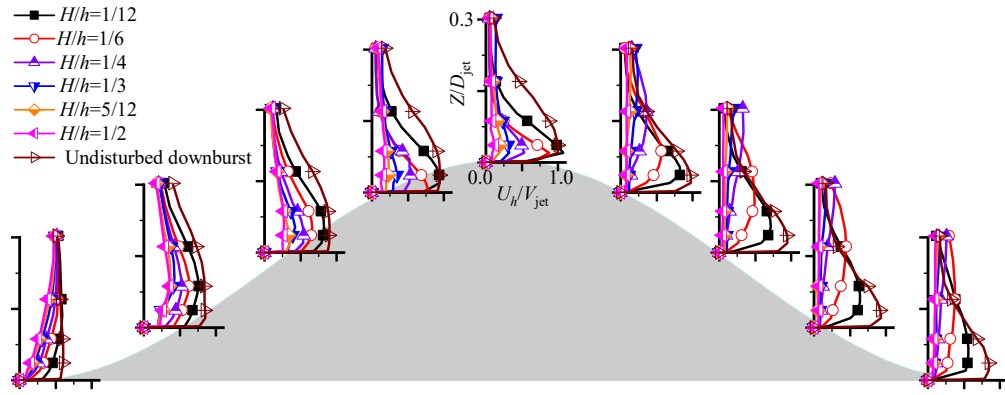

**Figure 7.** Profiles of horizontal wind velocity on *x*-axis ridge ($L = 0.5D_{jet}$, $r_h = 1.0D_{jet}$).

### 3.1. Horizontal Wind Velocity

For illustration, the downburst wind flows over the hill models with $L = 0.5D_{jet}$ and $r_h = 1.0D_{jet}$ are first discussed. Figures 7–9 show the profiles of the horizontal velocity at different measuring locations. For the same location, the corresponding profiles in the undisturbed downburst wind field where the wall jet thickness $\delta$ is about $0.15D_{jet}$ are also depicted in the figures. It is evident that the horizontal wind velocity varies with the relative height from the ground/hill surface. For convenience, we herein define "the global maximum" as the highest value for the entire wind field over the hill, and "the maximum" as the highest value for the same measuring location simultaneously measured at the different relative heights from the hill surface.

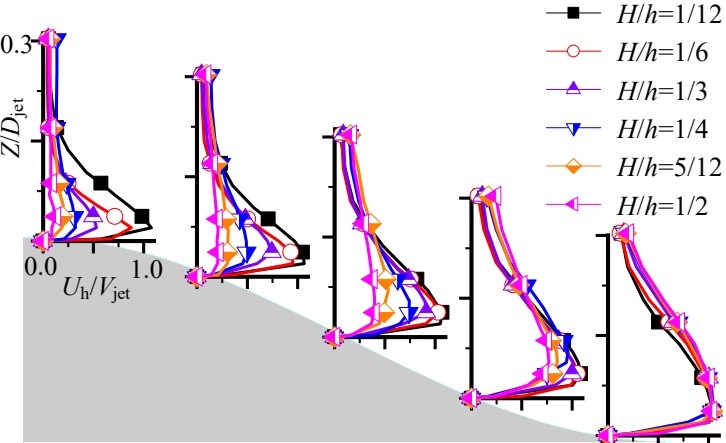

**Figure 8.** Profiles of horizontal wind velocity on *y*-axis ridge ($L = 0.5D_{\text{jet}}$, $r_{\text{h}} = 1.0D_{\text{jet}}$).

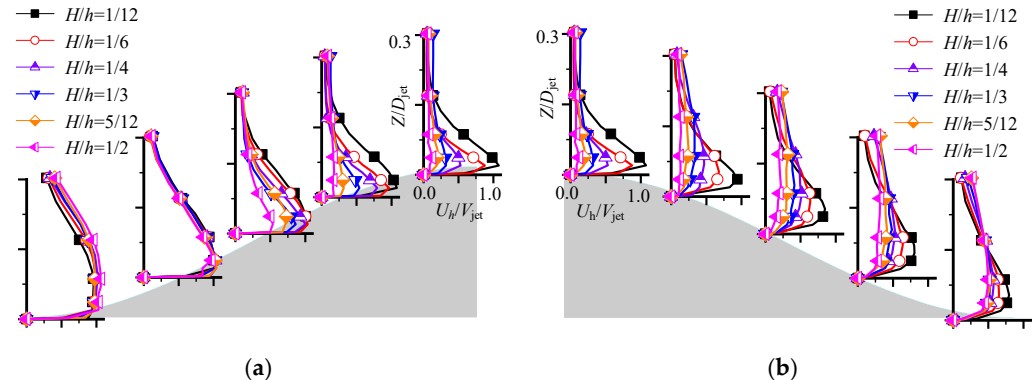

**Figure 9.** Profiles of horizontal wind velocity on 135°-ray (**a**) and 45°-ray (**b**) ridges ($L = 0.5D_{\text{jet}}$, $r_{\text{h}} = 1.0D_{\text{jet}}$).

Regarding the horizontal wind on the *x*-axis ridge, as shown in Figure 7, it was found that the effects of the hill with a low height ($H/h = 1/12$) were relatively small in terms of the shape of the wind profile, as well as the location where the maximum horizontal wind velocity occurred (around $1.0D_{\text{jet}}$). The predominant factor in determining the variation in the wind profile was the distance from the stagnation point. However, provided $H/h > 1/12$, the increase in $H/h$ was found to reduce the intensity of the horizontal wind in general. It was also observed that the near-ground maximum horizontal wind velocity decreased correspondingly with the increase in the hill height. In the cases of $H/h > 1/3$, the wind velocities measured at the crest were even lower than those measured near the windward hill foot. This may be attributed to the blockage effect caused by the hill becoming increasingly significant as the height of the hill increased, weakening the intensity of the downburst wind flow over the crest. The speed-up phenomenon on the *x*-axis ridge was only observed in the case of the hill with low elevation ($H/h = 1/12$), and occurred near the crest of hill, with a corresponding $M_{\text{t,max}}$ of $1.07V_{\text{jet}}$. Furthermore, it was found that the horizontal wind velocities measured on the windward side were generally greater than those measured on the leeward part.

The wind profiles on the *y*-axis ridge are shown in Figure 8. At the hill foot with a radial distance of roughly $r = 1.12D_{\text{jet}}$, the wind intensity became relatively stronger. The maximum wind velocity at the hill foot was a little greater than the impinging jet velocity. However, regardless of the hill height, the wind profiles measured at the hill foot were generally similar, which implies that the hill height had a negligible effect at this location. For the hills with $H/h > 1/6$, it was found that the horizontal wind velocity was significantly decreased along the ridge, implying that the blockage effect was pronounced. Furthermore, corresponding with the increase in the hill height, the maximum wind velocity decreased.

If $H/h \leq 1/6$, the speed-up phenomenon could be observed along the whole ridge, and the maximum topographic multiplier was found at $z = H/4$ with a value of $1.08V_{jet}$.

In Figure 9, the profiles of the horizontal wind velocity on the 45°-ray ridge (leeward side), as well as that on the 135°-axis ridge (windward side), are depicted. Generally, the wind velocities on the 135°-axis ridge were greater than those on the 45°-ray ridge. The profiles of horizontal wind velocity near the windward hill foot were similar, showing a negligible effect of hill height at that site. In addition, the corresponding wall jet thickness at that site was about $0.26D_{jet}$, which is greater than that recorded during an undisturbed downburst wind ($0.15D_{jet}$), indicating an increased wind intensity. In contrast, the wind profiles along the 45°-ray ridge, whether recorded at the hill foot or at the crest, were influenced by the hill height, which indicates that the blockage effect on the leeward wind field was relatively significant.

The contour maps of $M_{t,max}$ are depicted in Figure 10 and show the changes in wind intensity. As the hill height increases, the area of the speed-up region becomes increasingly small and the center of the region gradually moves to the windward hill foot lying on the 135° ray. In the cases of high hills, it was found that the speed-up regions when $M_{t,max} \geq 1$ were mainly located at the two sides of the hill. This again indicates that the downburst-induced near-ground strong airflow would be blocked if the hill is sufficiently high, and behaves like a flow around the hill body. In general, provided a high hill, the maximum horizontal wind velocity recorded for the crest was lower than that found at the hill foot, as shown in Figure 10. For a hill with a low height, e.g., the case of $H = h/12$, in which the near-ground wind is capable of climbing over the hill, the speed-up region comprised a relatively larger area in the vicinity of the crest. These findings are also corroborated by the distribution of the wind velocity vector over the hill (see Figure 11), where the arrows represent the directions and the magnitudes of the horizontal wind velocities. This flowing-around phenomenon was not observed in the previous studies using 2D hill models.

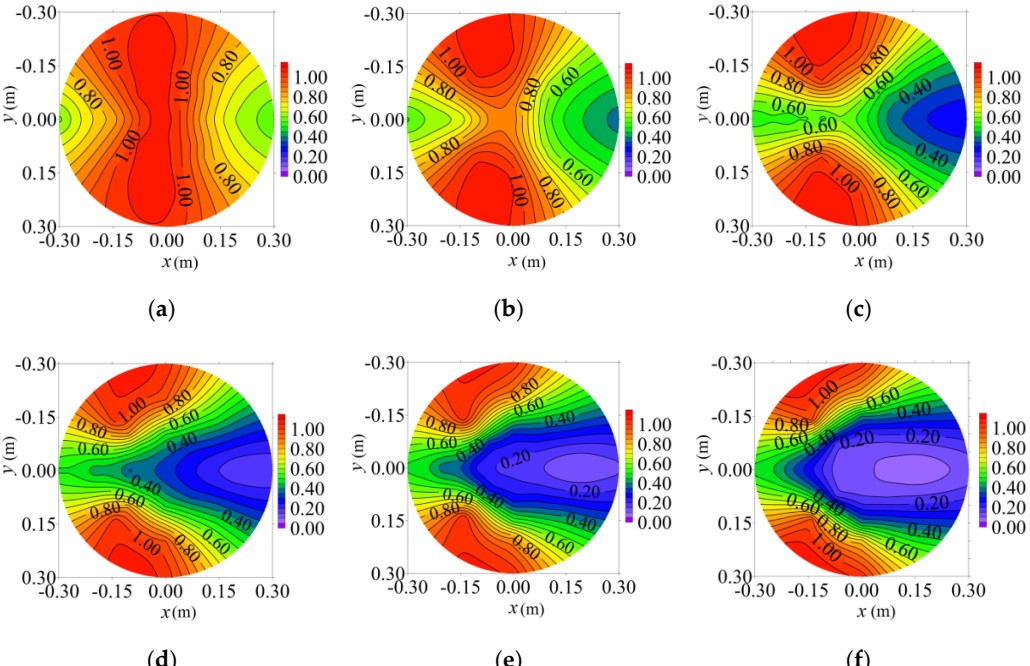

**Figure 10.** Contour maps of $M_{t,max}$ for horizontal wind velocity ($L = 0.5D_{jet}$, $r_h = 1.0D_{jet}$): (**a**) $H/h = 1/12$; (**b**) $H/h = 1/6$; (**c**) $H/h = 1/4$; (**d**) $H/h = 1/3$; (**e**) $H/h = 5/12$; (**f**) $H/h = 1/2$.

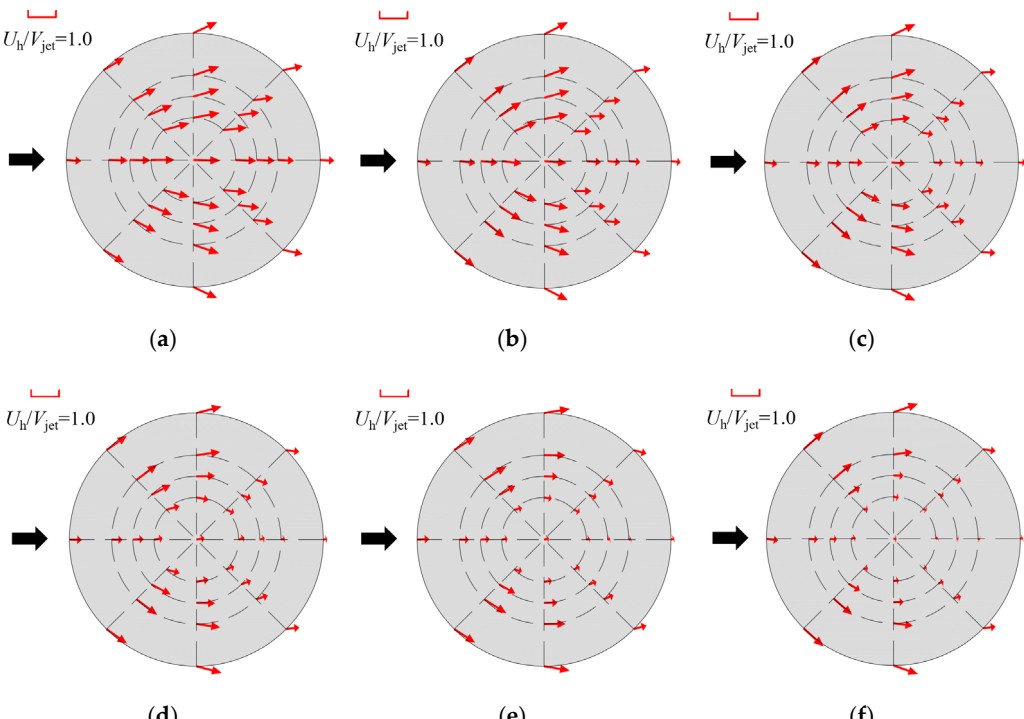

**Figure 11.** Vectors of the maximum horizontal wind velocities ($L = 0.5D_{jet}$, $r_h = 1.0D_{jet}$): (**a**) $H/h = 1/12$; (**b**) $H/h = 1/6$; (**c**) $H/h = 1/4$; (**d**) $H/h = 1/3$; (**e**) $H/h = 5/12$; (**f**) $H/h = 1/2$.

Figures 12 and 13 show the wind profiles when the hill models were placed at $r_h = 0.8D_{jet}$ and $r_h = 1.2D_{jet}$. The variation in the wind field was similar to the foregoing findings in the case of $r_h = 1.0D_{jet}$. That is, the profiles for the hill models at these three hill positions show that the speed-up regions were mainly located in the vicinity of the edge on the windward side, indicating that more attention should be paid to wind velocity in this region. In addition, provided $0.8D_{jet} \leq r_h \leq 1.2D_{jet}$, the location of the hill model had insignificant effects on the magnitude of the wind profile near the hill foot. This lack of impact of the hill position is consistent with the findings of work by Mason et al. [9,10]. Generally, the wind profiles in the vicinity of the crest, as well as those on the leeward side, were relatively more susceptible to the hill height.

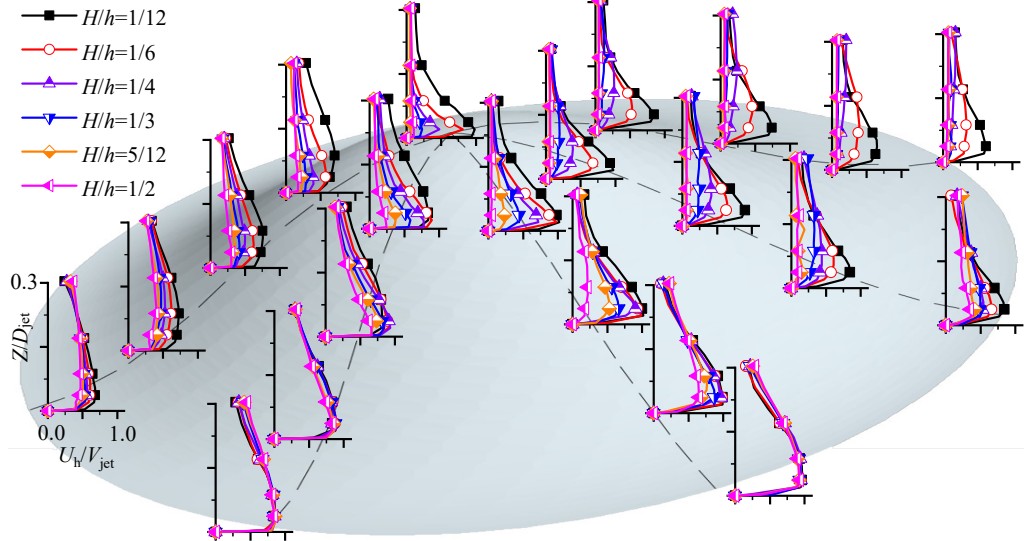

**Figure 12.** Profiles of horizontal wind velocity ($L = 0.5D_{jet}$, $r_h = 0.8D_{jet}$).

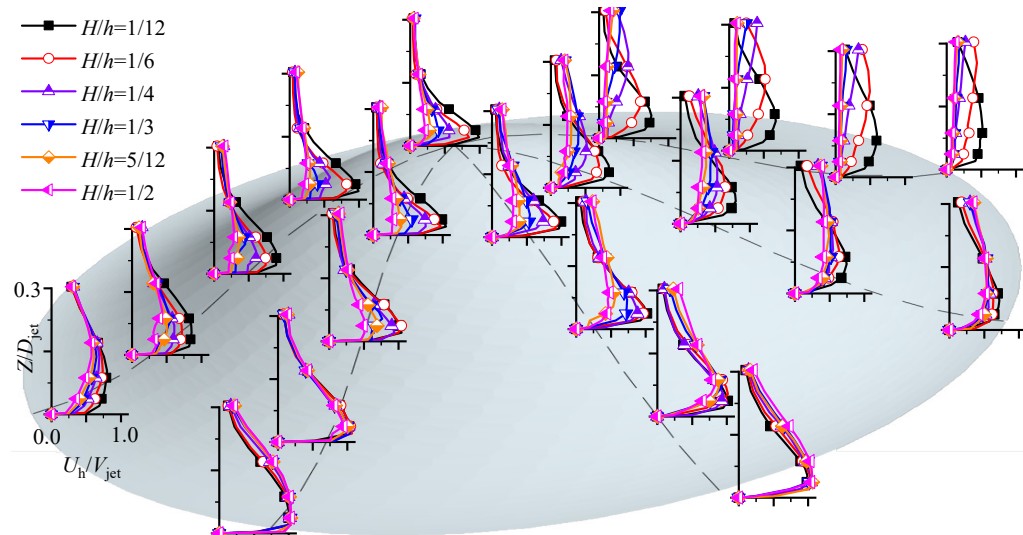

**Figure 13.** Profiles of horizontal wind velocity ($L = 0.5D_{jet}$, $r_h = 1.2D_{jet}$).

It is worth noting that in the cases surveyed, most of the profiles of the horizontal wind velocity have similar shapes to the downburst profile proposed by Wood et al. [8], i.e., correspondingly with the increase in the relative height from the ground/surface, the velocity increases almost linearly, then rapidly decreases after reaching a peak. Comparatively, the vertical profile of the horizontal wind in the ABL wind (ABL wind profile) usually exhibits an exponential/logarithmic shape. In the cases of high hills, it was found that the shape of the wind profile at the windward side of the *x*-axis ridge changed from a shape similar to an ABL wind profile to a shape more characteristic of a downburst profile. This phenomenon can be also observed in other cases, e.g., $L = 1.0D_{jet}$, as shown in Figures 14–16. Furthermore, the similarity to the ABL wind profile was usually but not always observed when $r/D_{jet}$ was around 0.5, and occasionally when $r/D_{jet} > 1.0$ for high hills ($H/h > 1/6$).

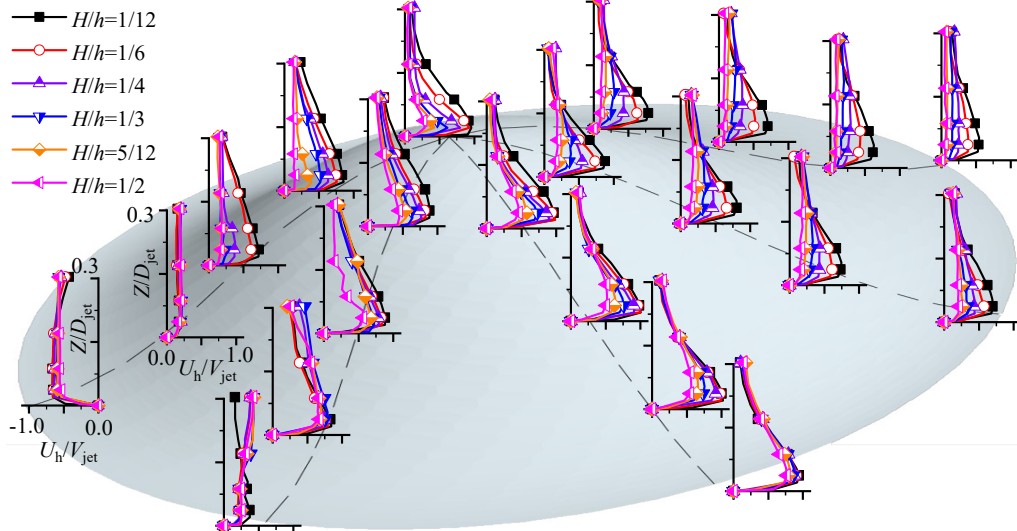

**Figure 14.** Profiles of horizontal wind velocity ($L = 1.0D_{jet}$, $r_h = 0.8D_{jet}$).

Figure 17 shows the contour maps of the global maximum horizontal velocity, where the conditions under which the speed-up phenomenon occurred can be clearly identified. In the figure, the abscissa represents the hill position, and the ordinate is the ratio of hill height to jet height. The results show that the maximum horizontal velocity occurred when $r_h$ was around $1.0D_{jet}$. In the cases of the hills with $L = 0.5D_{jet}$, it was found that a lower

hill commonly resulted in a higher wind speed. However, in the case of the hill models with $L = 1.0D_{jet}$, the maximum of $M_{t,max}$ was found to be 1.12 when $H/h = 5/12$.

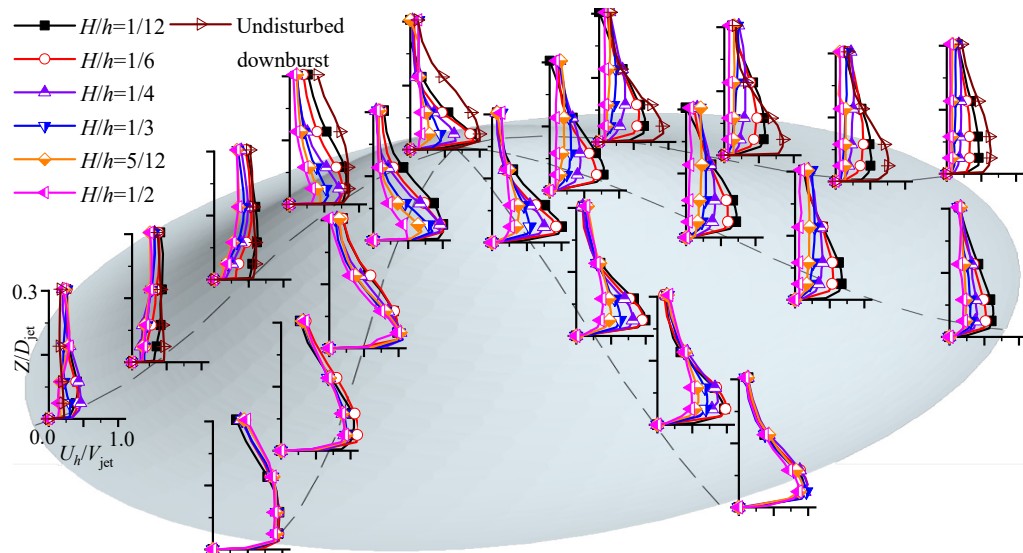

**Figure 15.** Profiles of horizontal wind velocity ($L = 1.0D_{jet}$, $r_h = 1.0D_{jet}$).

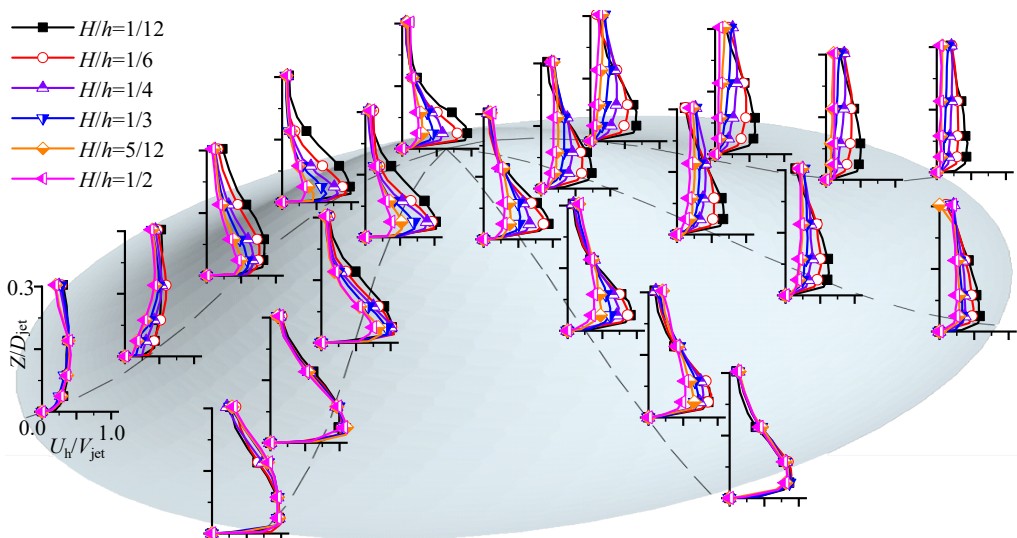

**Figure 16.** Profiles of horizontal wind velocity ($L = 1.0D_{jet}$, $r_h = 1.2D_{jet}$).

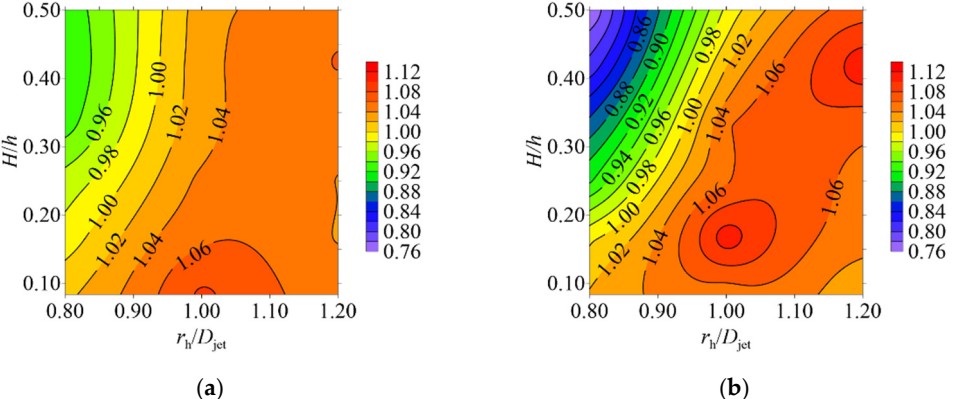

**Figure 17.** Contour maps of the global maximum horizontal velocity: (**a**) $L = 0.5D_{jet}$; (**b**) $L = 1.0D_{jet}$.

In order to further investigate where a speed-up wind would occur on the hilly terrain, a relative altitude, denoted as $Z_s$, was employed. The relative altitude was designated as the relative height from the hill surface to $Z_0$, the mean altitude of the hilly terrain in the stagnation region, which may be expressed as $Z_s = z - Z_0$. For each hill model, the maximum for the relative altitudes in the speed-up region, denoted as $Z_{s,max}$, was identified and associated with a corresponding radial distance of $r_{s,max}$ from the stagnation point. Figure 18 shows $Z_{s,max}$ versus $r_{s,max}$. It was found that the average of $Z_{s,max}$ was about $0.166D_{jet}$ ($1.1\delta$). This indicates that the speed-up would occur at the site with a relative altitude of around/below $0.166D_{jet}$, resulting in a more severe downburst wind.

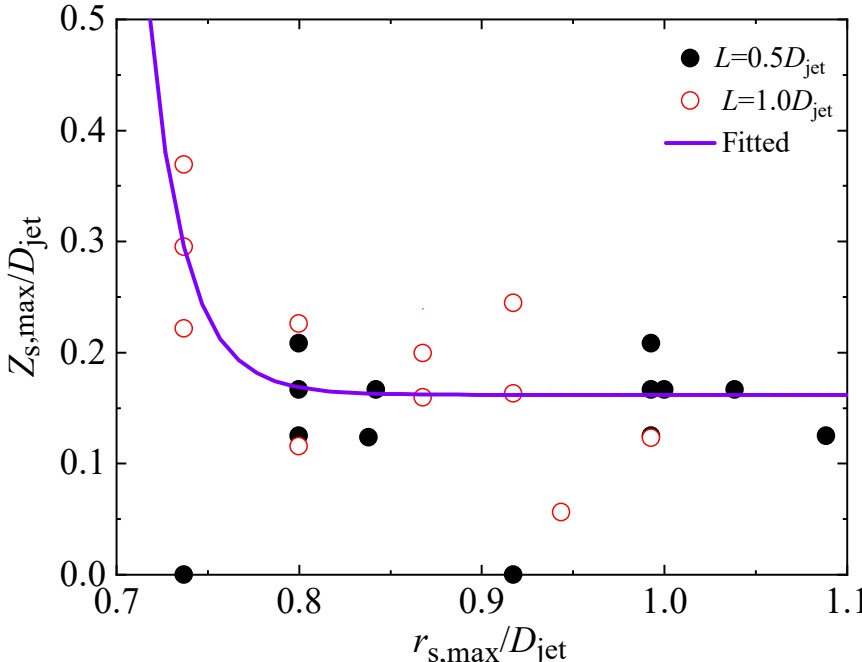

**Figure 18.** $Z_{s,max}$ versus $r_{s,max}$.

### 3.2. Vertical Wind Velocity

For illustration, Figure 19 shows the corresponding profiles obtained experimentally for vertical wind velocity for the hill models with $L = 0.5D_{jet}$ and $r_h = 1.0D_{jet}$. In the figure, a negative value indicates a downward flow. It was found that the vertical wind velocities were lower than the horizontal wind velocities in general, and the global maximum of the former was less than $0.5V_{jet}$, except for the results measured at the hill foot lying on the $x$-axis ridge. Focusing on the $x$-axis ridge, the profile of vertical wind velocity varied with the altitude of the measuring site. On the windward part, the variation in wind direction with the relative height from the surface of the hill, identified in the same profile, indicates the occurrence of circulation on the ridge. This might be attributed to the flow hitting the hill surface and being forced into an upward flow to ascend the hill. In addition, it was found that the position of the circulation on the windward ridge gradually moved toward the hill foot, as the hill height increased. On the leeward part, a downward flow was observed.

Considering the 135°-ray ridge (windward), the profiles also show the occurrence of circulation on the ridge, which was relatively weaker compared to that on the $x$-axis ridge. In addition, the weak vertical wind found at the hill foot implies a flowing-over horizontal wind at that place. For the sites located at the $y$-axis and 45°-ray ridges, the wind profiles also showed a downward flow. Moreover, the vertical winds on the 135°-ray, $y$-axis and 45°-ray ridges were relatively weaker, compared to that found on the $x$-axis ridge.

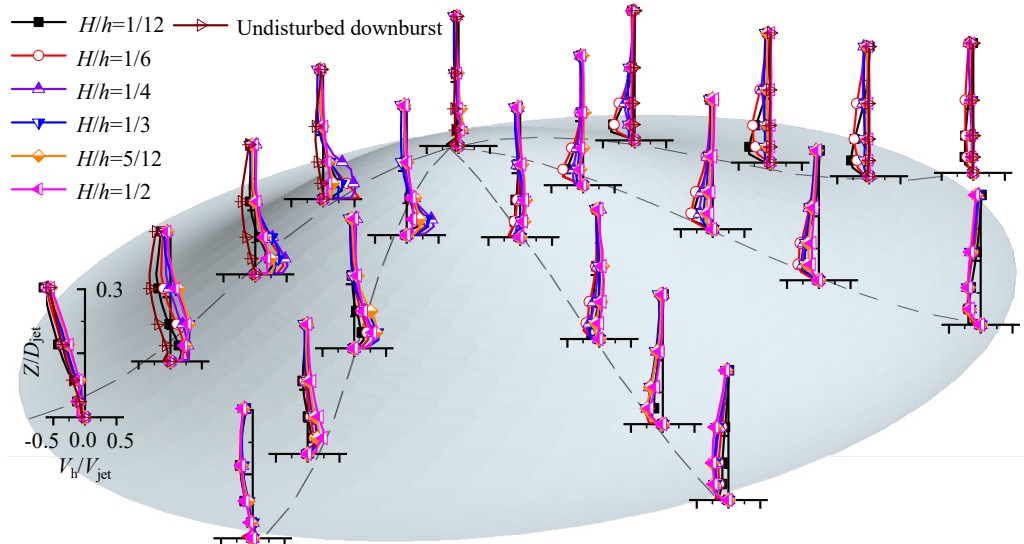

**Figure 19.** Profiles of vertical wind velocity ($L = 0.5D_{jet}$, $r_h = 1.0D_{jet}$).

Figure 20 shows the contour maps of the maximum vertical wind velocity. It was observed that both the global maximum vertical wind velocities corresponding to the upward airflow and those corresponding to the downward airflow initially increased with the increase in hill height before reaching a peak value, then decreasing. The peak value for the upward airflow was $0.5V_{jet}$ ($H/h = 1/4$), whereas that for the downward flow was $0.65V_{jet}$ ($H/h = 1/6$). Regarding the downward wind, it was found that the global maximum typically occurred at the leeward part of the $x$-axis ridge if $H/h \leq 1/6$, and at the hill foot of the $y$-axis ridge if $H/h > 1/6$. With regards to the upward wind, the global maximum appeared at the windward $x$-axis ridge if $H/h \leq 1/4$, and at the two sides of the hill if $H/h > 1/4$. These findings again highlight that if the hill is sufficiently high, a circumfluent flow occurs on the sides of the hill.

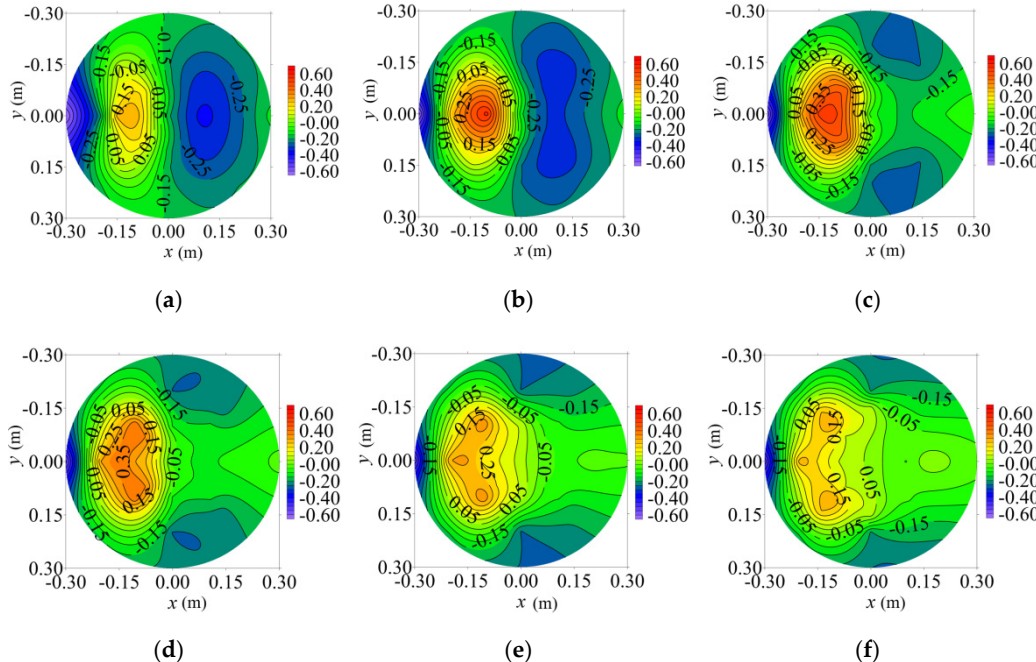

**Figure 20.** Contour maps of the maximum vertical wind velocity ($L = 0.5D_{jet}$, $r_h = 1.0D_{jet}$): (**a**) $H/h = 1/12$; (**b**) $H/h = 1/6$; (**c**) $H/h = 1/4$; (**d**) $H/h = 1/3$; (**e**) $H/h = 5/12$; (**f**) $H/h = 1/2$.

To clarify the impact of the hill position on the maximum vertical wind velocity, Figure 21 illustrates the corresponding contour maps for a typical hill model ($L = 0.5D_{jet}$ and $H/h = 0.25$) placed at $r_h = 0.8D_{jet}$, $r_h = 1.0D_{jet}$, and $r_h = 1.2D_{jet}$, respectively. These locations of the hill models are around where the strongest near-ground horizontal wind would be observed in a corresponding undisturbed downburst. The three subplots show a similar phenomenon; that is, the upward airflow occurs on the windward side and the downward airflow mainly occurs on leeward side. In addition, correspondingly with the increase in the horizontal distance from the downburst, the vertical wind velocity of the upward flow increased, whereas that of the downward flow decreased. Furthermore, the strong upward flow mainly appeared around the *x*-axis ridge, whereas the strong downward flow appeared at the hill foot of the *y*-axis ridges.

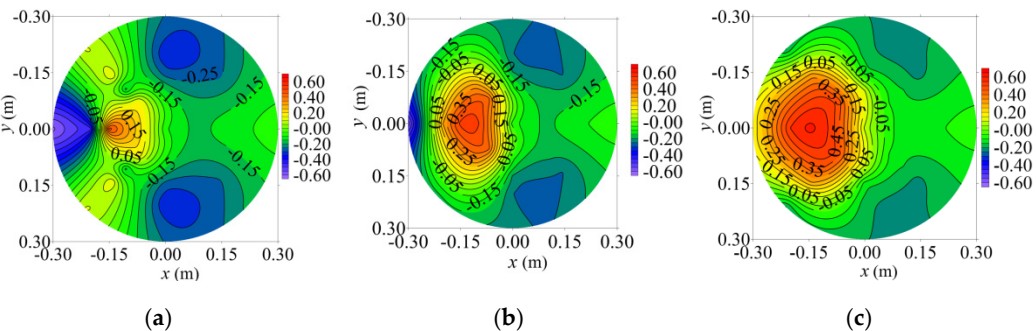

**Figure 21.** Contour maps of the maximum vertical wind velocity ($L = 0.5D_{jet}$, $H/h = 1/4$): (**a**) $r_h = 0.8D_{jet}$; (**b**) $r_h = 1.0D_{jet}$; (**c**) $r_h = 1.2D_{jet}$.

Though the foregoing analyses show that a significant vertical wind field was observed when the hill model was placed in the strong horizontal wind region, it is a priori true that the vertical wind field above the hill would be much stronger when the hill model is placed in the stagnation region. Figure 22 shows the profiles of vertical wind velocity when the hill models were located at $r_h = 0.0D_{jet}$, namely directly below the impinging jet. The maximum among the measured vertical wind velocities at the crest was about $1.0V_{jet}$. Comparison of the results with the wind profiles at the same locations in the downburst over a flat surface shows a similar curve shape. Generally, the vertical wind velocity at a site that was farther from the crest/stagnation point was lower. In the stagnation region, the vertical wind velocities increased monotonically with the increase in the relative height from the hill surface, as well as the increase in the hill height, which implies that the variation in wind velocity corresponding to height can be mainly attributed to the vertical distance from the nozzle; that is, a shorter vertical distance results in a higher vertical wind velocity.

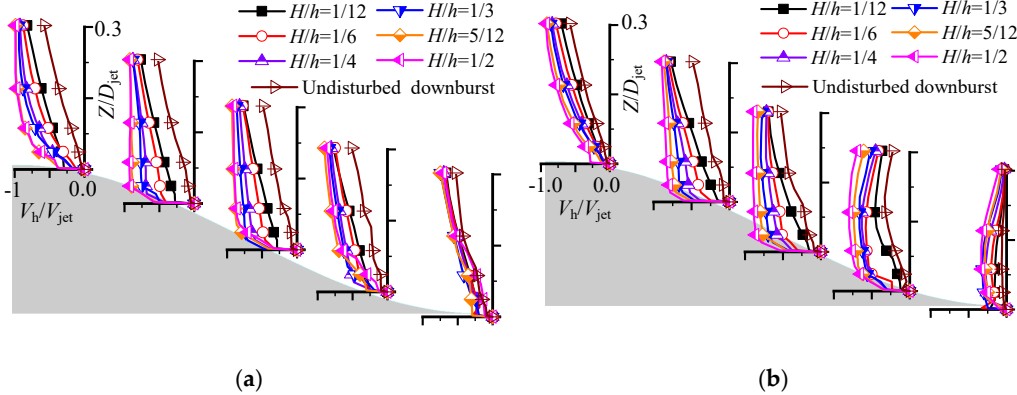

**Figure 22.** Profiles of vertical wind velocity ($r_h = 0.0D_{jet}$): (**a**) $L = 0.5D_{jet}$; (**b**) $L = 1.0D_{jet}$.

Figure 23 shows the contour maps of the vertical velocity at the crest. Note that the hill model in Figure 23a has a steeper slope than the hill model in Figure 23b, but the two

hill models have the same hill height. It was found that a steeper slope led to a greater gradient in the vertical velocity.

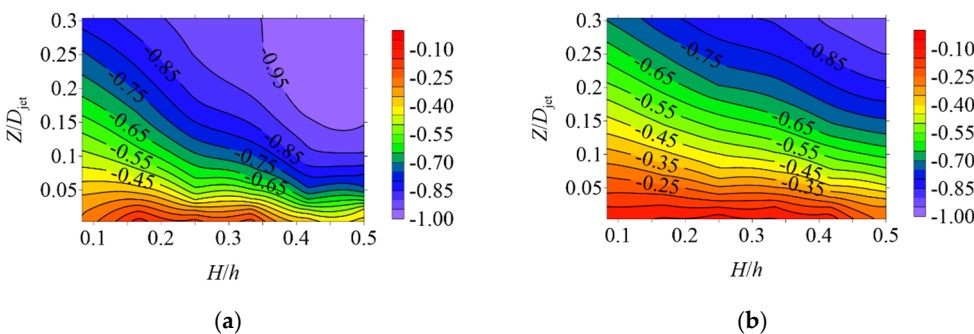

**Figure 23.** Contour maps of vertical wind velocity ($r_h = 0.0D_{jet}$): (**a**) $L = 0.5D_{jet}$; (**b**) $L = 1.0D_{jet}$.

### 3.3. Comparison with the Results of Previous Studies

Since the results of previous studies mainly pertain to horizontal wind dynamics, this section focuses only on the comparison of the horizontal wind velocities. Regarding the hill models located in the strong horizontal wind region, the maximum $M_{t,max}$ for each model and hill position are summarized in Table 4. The maximum $M_{t,max}$ shown in the table could aid in determining downburst wind loads and inform a simplified design of structures subjected to such conditions. It is evident that both the maximum $M_{t,max}$ obtained at the crest and that observed at the hill foot decreased with the increase in the hill height. In addition, the maximums of $M_{t,max}$ obtained elsewhere were relatively large, in the range of 1.03 to 1.12. Note that the work by Shen et al. [22], utilizing a hill model with a similar shape to those employed in the present study, found that the maximum speed-up ratio during an ABL wind flow over a 3D hill was about 1.34. It can be thus concluded that the speed-up effect found during a downburst wind is commonly smaller than occurs during an ABL wind, provided the hill terrain is the same. This is consistent with the findings of previous studies [6,7].

**Table 4.** Maximum of $M_{t,max}$ for hill models.

| Location | Hill Models | | | | | | | | | | | |
|---|---|---|---|---|---|---|---|---|---|---|---|---|
| | **H1** | **H2** | **H3** | **H4** | **H5** | **H6** | **H7** | **H8** | **H9** | **H10** | **H11** | **H12** |
| Crest | 1.07 | 0.88 | 0.56 | 0.41 | 0.28 | 0.21 | 0.98 | 0.92 | 0.66 | 0.44 | 0.30 | 0.23 |
| Hill foot | 1.04 | 1.04 | 1.04 | 1.04 | 1.06 | 1.06 | 0.97 | 0.98 | 0.96 | 0.96 | 0.96 | 0.95 |
| Else | 1.08 | 1.07 | 1.03 | 1.03 | 1.04 | 0.97 | 1.05 | 1.11 | 1.06 | 1.06 | 1.12 | 1.07 |

The Australian and New Zealand standard entitled "Overhead line design—detailed procedures" (AS/NZS 7000: 2010) [29] specifies that the topographical multiplier $M_{t,downdraft}$ for downbursts can be computed using the formula

$$M_{t,\,downbnurst} = 0.5 + 0.5 M_{t,\,synoptic} \tag{3}$$

where $M_{t,synoptic}$ is the topographical multiplier for synoptic winds and is defined in the code entitled "Structural design actions-Part 2: wind actions" (AS/NZS 1170.2: 2011) [30]. For the present study, $M_{t,synoptic} = M_h$, where $M_h$ is the hill shape multiplier specified in AS/NZS 1170.2: 2011 [30] and can be computed using the formula

$$M_h = \begin{cases} 1, & \phi < 0.05 \\ 1 + \left(\frac{H}{3.5(Z+L_1)}\right)\left(1 - \frac{|x|}{L_2}\right), & 0.05 \le \phi \le 0.45 \\ 1 + 0.71\left(1 - \frac{|x|}{L_2}\right), & \phi > 0.45 \text{ and } x \in [0, \frac{H}{4}] \\ 1 + \left(\frac{H}{3.5(Z+L_1)}\right)\left(1 - \frac{|x|}{L_2}\right), & \phi > 0.45 \text{ and } x \notin [0, \frac{H}{4}] \end{cases} \tag{4}$$

where $L_1$ and $L_2 = 4L_1$ are the length scales used to determine the vertical and the horizontal variations in $M_h$, and $L_1$ can be taken as the greater of $0.36L_u$ or $0.4H$.

For the $M_{t,max}$ at the crest, Figure 24 shows the results from the previous research, the curves defined using Equation (3), and the experimental results from the present study. It can be observed that in the case of a small slope ($\phi \leq 0.45$) the results show good agreement. With the increase in the slope, the $M_{t,max}$ at the crest decreases, whereas AS/NZS 7000: 2010 [29] merely provides a constant topographic multiplier, which results in an excessively conservative design in general. There is a need for formulas that enable a relatively accurate prediction of $M_{t,max}$ in order to inform a rational design of structures. The discrepancy is largely attributable to the confined nature of the downburst. Moreover, the topographic multiplier defined for an ABL wind, e.g., Equation (4), is not suitable for a downburst wind, particularly in the case of high hills. Regarding the downburst wind, it is worth noting that a relatively higher $M_{t,max}$ than those illustrated in Figure 24 probably occurs at other locations, as shown in Table 4.

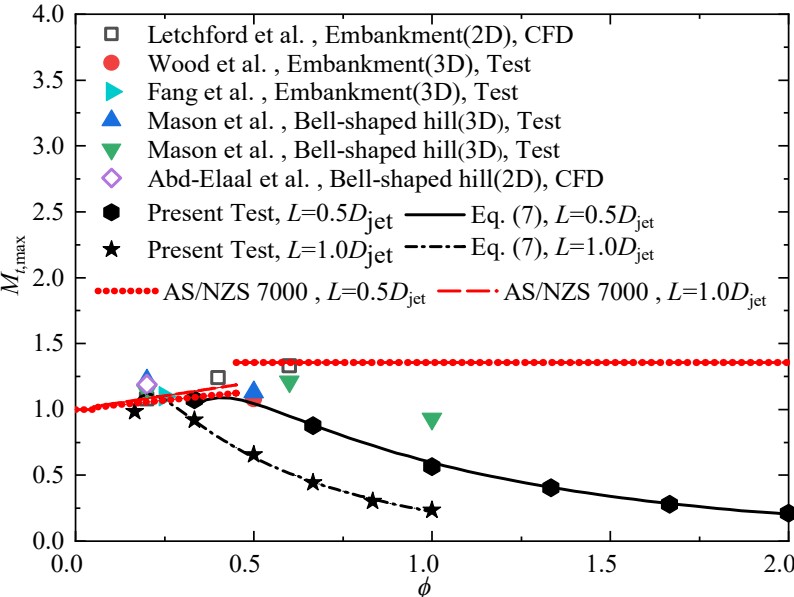

**Figure 24.** $M_{t,max}$ at crest.

## 4. Empirical Equations for Wind Profiles

### 4.1. Horizontal Wind

As the foregoing analyses show that the most significant speed-up occurs when $r_h = 1.0D_{jet}$, the corresponding cases were used to inform the determination of the design wind loads. Five positions, including the crest, the hill foot, and the locations with altitudes of $0.25H$, $0.5H$, and $0.75H$, respectively, are considered in this section. From the perspective of design, a conservative but reasonable wind load is preferred. $U_{max}$ denotes the maximum horizontal wind velocity of the profile. Thus, for each position at a given altitude in each case, the envelope values of horizontal wind velocity, in which the value at a relative height is the maximum among the corresponding horizontal velocities measured at all measuring locations, are utilized to enable a conservative design, as illustrated in Figure 25. In the figure, the normalized envelope values are compared with the curve proposed by Wood et al. [8] in the form of

$$\frac{U(Z)}{U_{max}} = 1.55(\frac{Z}{\delta})^{\frac{1}{6}} \left[1 - erf(0.7(\frac{Z}{\delta}))\right] \tag{5}$$

where $\delta$ is the wall jet thickness equal to the relative height from the ground to where the horizontal wind velocity is $0.5U_{max}$ [8]. As shown in Figure 25, except for the values

measured at the hill foot, the test results are in good agreement with Equation (5). Furthermore, based on Equation (5), a modified phenomenological model is summarized herein to characterize the test results measured at the hill foot, and has the form of

$$\frac{U(Z)}{U_{\max}} = 7.311(\frac{Z}{\delta})^{0.528}[1 - \text{erf}(-1.267(\frac{Z}{\delta}))] - 15.062\left(\frac{Z}{\delta}\right) + 1.475\left(\frac{Z}{\delta}\right)^3 \qquad (6)$$

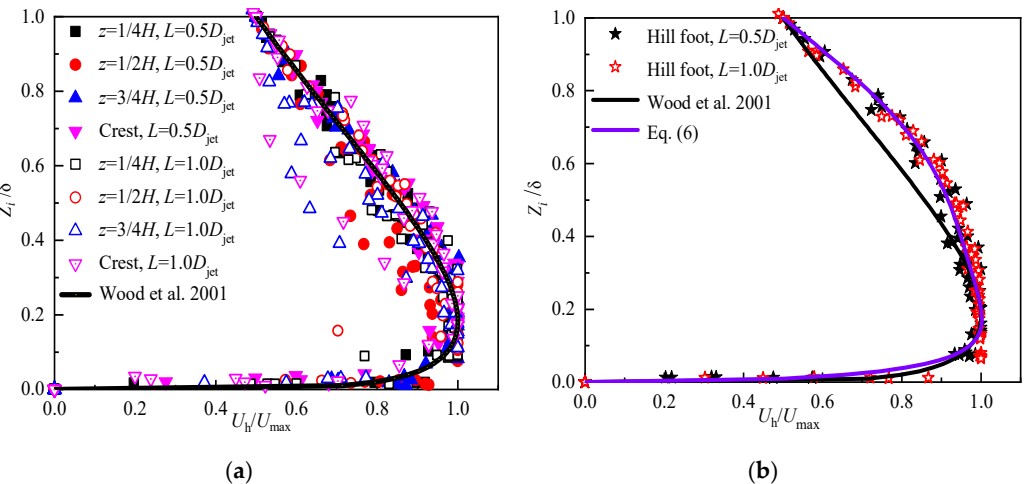

**Figure 25.** Normalized envelope values of horizontal wind velocity: (**a**) on hill; (**b**) hill foot.

The comparison of Equation (6) with the test results for the hill foot shows a satisfactory agreement, as shown in Figure 25b.

In Equations (5) and (6), $U_{\max}$ and $\delta$ can be estimated using the formulae

$$\delta/D_{\text{jet}} = a_1 + a_2\phi + a_3\phi^2 + a_4\phi^3 + a_5\phi^4 + a_6/\eta + a_7\eta + a_8\eta^2 + a_9\eta^3 \qquad (7a)$$

$$U_{\max}/V_{\text{jet}} = b_1 + b_2\phi + b_3\phi^2 + b_4\phi^3 + b_5/\eta^2 + b_6/\eta + b_7\eta + b_8\eta^2 + b_9\eta^3 + b_{10}\eta^4 \qquad (7b)$$

where $\eta = H/D_{\text{jet}}$. Moreover, $a_i$ ($i = 1, \ldots, 9$) and $b_i$ ($i = 1, \ldots, 10$) are the coefficients, and can be obtained by fitting the test results. The values of $a_i$ and $b_i$ are tabulated in Tables 5 and 6, respectively. Equation (7) accounts for the effects of the slope of the hill and the hill height on the wind profiles. Actually, considering that the maximum horizontal velocity observed in an undisturbed downburst wind field is approximately $1.0V_{\text{jet}}$, Equation (7b) provides a relatively accurate estimation of $M_{\text{t,max}}$, which would enable a rational and economic structure design. The comparison of the test results with the estimated results is illustrated in Figure 26. In the figure, selected hill models with $L = 0.5D_{\text{jet}}$ and selected hill models with $L = 1.0D_{\text{jet}}$ are used for the comparison. A satisfactory agreement is observed.

**Table 5.** Values of $a_i$.

|  | $a_1$ | $a_2$ | $a_3$ | $a_4$ | $a_5$ | $a_6$ | $a_7$ | $a_8$ | $a_9$ |
|---|---|---|---|---|---|---|---|---|---|
| Crest | 0.17 | −0.15 | 0.07 | 0 | 0 | 0 | −0.39 | 1.19 | −0.76 |
| $z = 3/4H$ | 0.24 | −0.56 | 0.54 | −0.13 | 0 | 0 | −0.07 | 0 | 0 |
| $z = 2/4H$ | 0.26 | −0.16 | −0.16 | 0.28 | −0.08 | −0.007 | 0 | 0 | 0 |
| $z = 1/4H$ | 0.31 | −0.28 | 0.14 | −0.02 | 0 | −0.01 | 0.09 | 0 | 0 |
| Hill foot | 0.20 | −0.20 | 0.13 | −0.03 | 0 | 0 | 0.59 | −0.78 | 0.39 |

The value $\gamma_U$ can be calculated as the ratio of the measured horizontal wind velocity, $U_{\text{Exp}}$, to the estimated horizontal wind velocity, $U_E$, as follows:

$$\gamma_U = U_{\text{Exp}}/U_E \qquad (8)$$

For velocities greater than 3 m/s, and in the range of $0.5U_{max}$ to $1.0U_{max}$, the mean value and the coefficient of variation (COV) of $\gamma_U$ are listed in Table 7. It can be observed that the COVs are less than 9%, indicating a relatively satisfactory agreement.

**Table 6.** Values of $b_i$.

|  | $b_1$ | $b_2$ | $b_3$ | $b_4$ | $b_5$ | $b_6$ | $b_7$ | $b_8$ | $b_9$ | $b_{10}$ |
|---|---|---|---|---|---|---|---|---|---|---|
| Crest | −0.16 | −0.14 | 0.04 | 0 | −0.056 | 0.54 | 0 | 0 | 0 | 0 |
| $z = 3/4H$ | 0.99 | 0.90 | −1.19 | 0.35 | 0 | 0 | −0.52 | 0 | 0 | 0 |
| $z = 2/4H$ | 1.01 | 0.19 | −0.22 | 0 | 0 | 0 | 0.007 | 0 | 0 | 0 |
| $z = 1/4H$ | 0.76 | 0. 01 | −0.04 | 0 | 0 | 0 | 3.31 | −11 | 14.31 | −6.28 |
| Hill foot | 1.01 | 0.43 | −0.24 | 0.06 | 0 | 0 | −0.64 | 0.58 | −0.24 | 0 |

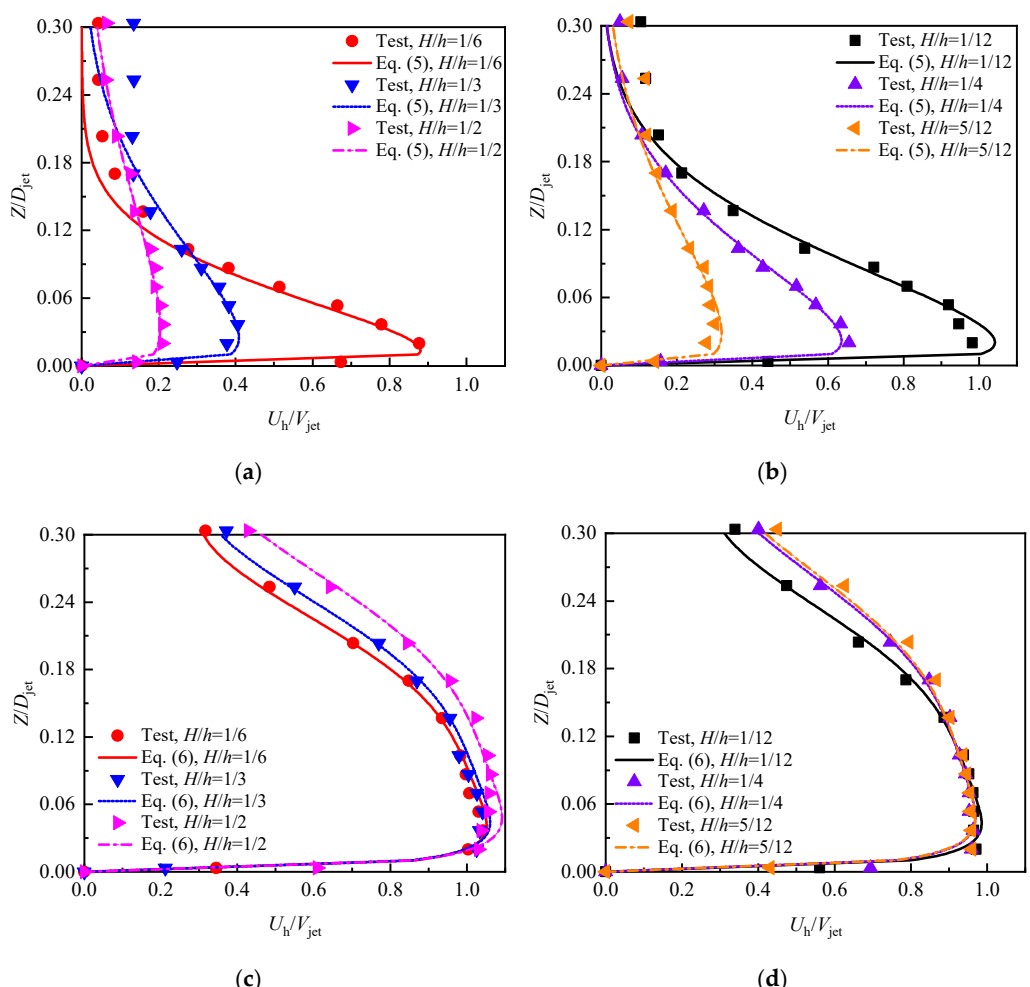

(a)　　　　　　　　　　　　　(b)

(c)　　　　　　　　　　　　　(d)

**Figure 26.** Experimental and predicted envelope values for horizontal wind velocity: (**a**) Crest, $L = 0.5D_{jet}$; (**b**) Crest, $L = 1.0D_{jet}$; (**c**) Hill foot, $L = 0.5D_{jet}$; (**d**) Hill foot, $L = 1.0D_{jet}$.

**Table 7.** Mean value and COV of $\gamma_U$.

|  | **Mean** | **COV** |
|---|---|---|
| Crest | 1.005 | 4.8% |
| $z = 3/4H$ | 0.976 | 8.7% |
| $z = 2/4H$ | 0.985 | 5.0% |
| $z = 1/4H$ | 0.982 | 4.4% |
| Hill foot | 0.996 | 2.5% |

### 4.2. Vertical Wind

Since the maximum vertical velocity occurs when $r_h = 0.0D_{jet}$, and is located at the crest, only the phenomenological model for the case of $r_h = 0.0D_{jet}$ was utilized for predictions of the vertical wind velocities at the crest, to ensure a conservative design. This model is in the form of

$$\frac{V_C(Z)}{V_{jet}} = \frac{p}{1 + q\frac{D_{jet}}{Z}} \tag{9}$$

where $V_C$, as a function of the relative height $Z$, is the vertical wind velocity on the crest, and $p$ and $q$ are the coefficients, which can be computed as follows:

$$p = c_1 + c_2\phi + c_3\phi^2 + c_4\phi^3 + c_5\phi^4 + c_6\eta$$
$$q = d_1 + d_2\phi + d_3\phi^2 + d_4\phi^3 + d_5\phi^4 + d_6\eta \tag{10}$$

where $c_i$ ($i = 1, \ldots, 5$) and $d_i$ ($i = 1, \ldots, 5$) are the coefficients, and can be obtained by fitting the test results. The values of $c_i$ and $d_i$ are tabulated in Table 8. Both the experimental results and the predicted results are illustrated in Figure 27, indicating a good agreement between them.

**Table 8.** Values of $c_i$ and $d_i$.

| $c_1$ | $c_2$ | $c_3$ | $c_4$ | $c_5$ | $c_6$ | $d_1$ | $d_2$ | $d_3$ | $d_4$ | $d_5$ | $d_6$ |
|-------|-------|-------|-------|-------|-------|-------|-------|-------|-------|-------|-------|
| 1.454 | −1.182 | 1.226 | −0.6082 | 0.1118 | 0.1118 | 0.3935 | −0.9798 | 0.9711 | −0.4579 | 0.08016 | 0.08016 |

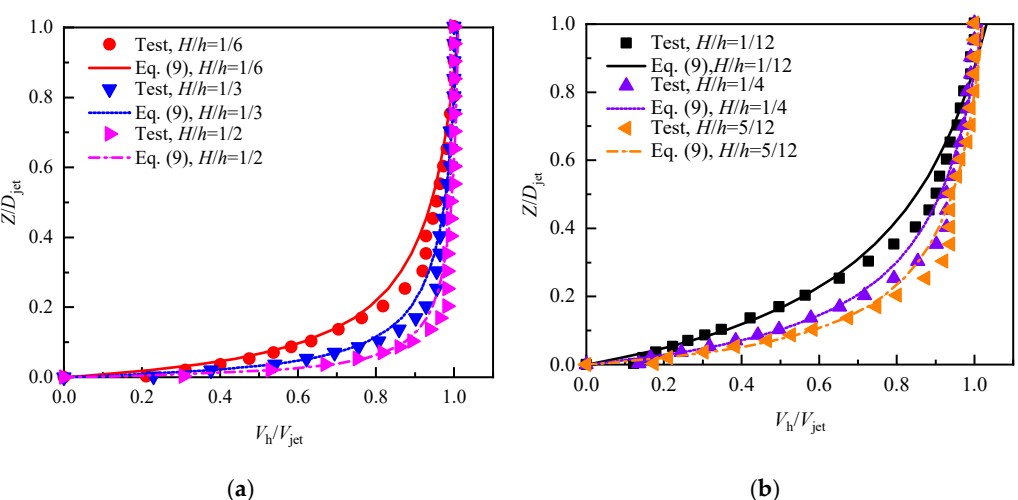

**Figure 27.** Experimental and predicted vertical wind velocities on crest ($r_h = 0.0D_{jet}$). (**a**) $L = 0.5D_{jet}$; (**b**) $L = 1.0D_{jet}$.

In a similar fashion, $\gamma_V$ is defined as the ratio of the measured vertical wind velocity, $V_{Exp}$, to the predicted vertical wind velocity, $V_E$, as follows:

$$\gamma_V = V_{Exp}/V_E \tag{11}$$

With regards to the results measured at relative heights greater than $0.05D_{jet}$, the mean value and the COV of $\gamma_W$ are 1.012 and 2.6%, respectively.

### 5. Conclusions

1.  The laboratory tests conducted using a physical downburst simulator revealed remarkable effects of hilly terrain on the downburst wind field. It was found that the speed-up was not only related to the slope, but also to the hill height and ratio of hill

height to jet height. This finding is evidently different from the ABL wind case, and can be largely attributed to the nature of the downburst.

2.  Horizontal wind velocity predominated when the hill was placed in the strong horizontal wind region, i.e., when the distance from the stagnation point to the crest was around $1.0D_{jet}$. Under this circumstance, the wind tended to flow over the crest when the height of the hill was sufficiently low, and flowed around the hill body in the cases of high hills. As the hill height increased, the speed-up region gradually moved from the crest to the sides of the hill, namely to the vicinity of the windward edges. As a result, the maximum speed-up effect was frequently found on the ridge or at the hill foot, in the cases of high hills. Among all cases, the maximum topographic multiplier was 1.12, and appeared at the ridge when the ratio of hill height to jet height was 5/12, in terms of horizontal wind. In addition, the corresponding maximum vertical wind velocity value reached about half that of the impinging jet velocity.

3.  A strong vertical wind field on the hill was observed when the hill model was placed directly under the impinging jet. It was found that the maximum strong vertical wind appeared at the crest of the hill, and that either a steeper slope or a higher hill led to a greater gradient of vertical wind velocity in the direction of elevation. As the hill height increased, the vertical wind velocity at the relative height of $0.3D_{jet}$ would increasingly approach the impinging jet velocity.

4.  To inform future structure design, explicit formulas for estimating the envelope values at the typical locations were presented that enable conservative estimations of horizontal downburst wind loads. Moreover, a phenomenological model was proposed to predict the vertical wind velocities on the crest when the hill is directly under the impinging jet, and shows a satisfactory agreement with the test results.

5.  The physical downburst simulator employed herein is mainly based on the impinging jet model that has been widely accepted in wind engineering for the experimental study of downburst wind and its effects on structures. However, the simulator is not capable of reproducing the dynamic characteristics of the downburst, in particular the ring vortex that is occasionally observed during downburst events. Accordingly, there is a need for further study to examine the effects of the presence of ring vortices.

**Author Contributions:** Conceptualization, Y.C. and G.S.; methodology, Y.C., G.S., Y.L. and W.L.; investigation, Y.L., J.Y., Y.G. and H.X.; writing—original draft preparation, Y.C. and Y.L.; writing—review and editing, Y.C., G.S. and H.X. All authors have read and agreed to the published version of the manuscript.

**Funding:** This research was funded by the National Natural Science Foundation of China.

**Institutional Review Board Statement:** Not applicable.

**Informed Consent Statement:** Not applicable.

**Data Availability Statement:** Not applicable.

**Acknowledgments:** The authors greatly appreciate the support of the National Natural Science Foundation of China under Grant No. 51878607, 51838012 and 52178511. The opinions and statements do not necessarily represent those of the sponsors.

**Conflicts of Interest:** The authors declare no conflict of interest.

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
