# Peer review of "Experimental Study of Downburst Wind Flow over a Typical Three-Dimensional Hill"

_applsci, doi:10.3390/app12063101_

Round 1

Reviewer 1 Report

Please see the attached document for reviewer comments.

Reviewer 2 Report

The authors provided a very good study supported by abundant data and analysis. This is a good job and should be published with the current version. 

Author Response

Thanks for the valuable comments on the manuscript.

Reviewer 3 Report

[In the abstract]

1) The authors give a specific description of the model, like ‘jet diameter of 600 mm’. The reviewer thinks that some statistical results can be given other than the present qualitative results.

[In introduction]

2) In the beginning of the introduction, it is suggested to indicate the implication of anti-wind research on the safety of various types of industrial infrastructure, such as the bridge [1], the transmission line [2], and the rail overhead line [3]. This can be a good motivation to introduce the main work of this paper.

[1] Hao, Jianming, and Teng Wu. "Downburst-induced transient response of a long-span bridge: A CFD-CSD-based hybrid approach." Journal of Wind Engineering and Industrial Aerodynamics 179 (2018): 273-286.

[2] Aboshosha, H.; Elawady, A.; El Damatty, A.; etc. Review on dynamic and quasi-static buffeting response of transmission lines under synoptic and non-synoptic winds. Eng. Struct. 2016, 112, pp.23-46.

[3]  Song, Y.; Zhang, M.; Øiseth, O.; Rønnquist, A. Wind deflection analysis of railway catenary under crosswind based on nonlinear finite element model and wind tunnel test. Mech. Mach. Theory 2022, 168.

3) The importance of accurately evaluating the maximum of Mt,max can be highlighted, as it is of the important motivations of this work.

[In section 2]

4) ‘as shown in Figure 2’ should be ‘as shown in Figure 2 (a)’ for more specific.

5) Some legends may be included in Figure 2 (a) for a better understanding of the test rig.

6) What is the material of the hill model? Is there any concerns of issues about the surface roughness and flexibility of the hill model?

[In section 3]

7) ‘Figure 12-13’ should be ‘Figures 12-13’.

8) As claimed by the authors in the introduction, what is the new finding of this study against previous ones by improving the experimental model. Please directly indicate this when analysing the results.

Round 2

Reviewer 3 Report

Thanks for the elaborate response to my comments.